# Exploring the Potential of Nanoporous Materials for Advancing Ophthalmic Treatments

**DOI:** 10.3390/ijms242115599

**Published:** 2023-10-26

**Authors:** Kevin Y. Wu, Danielle Brister, Paul Bélanger, Simon D. Tran

**Affiliations:** 1Department of Surgery—Division of Ophthalmology, University of Sherbrooke, Sherbrooke, QC J1G 2E8, Canada; yang.wu@usherbrooke.ca (K.Y.W.);; 2College of Public Health, National Taiwan University (NTU), Taipei 106319, Taiwan; 3Faculty of Dental Medicine and Oral Health Sciences, McGill University, Montreal, QC H3A 1G1, Canada

**Keywords:** nanoporous materials, ophthalmology, drug delivery systems, intraocular lenses, contact lenses, biocompatibility, nanotechnology, ocular diseases, material science, controlled release

## Abstract

The landscape of ophthalmology is undergoing significant transformations, driven by technological advancements and innovations in materials science. One of the advancements in this evolution is the application of nanoporous materials, endowed with unique physicochemical properties ideal for a variety of ophthalmological applications. Characterized by their high surface area, tunable porosity, and functional versatility, these materials have the potential to improve drug delivery systems and ocular devices. This review, anchored by a comprehensive literature focusing on studies published within the last five years, examines the applications of nanoporous materials in ocular drug delivery systems (DDS), contact lenses, and intraocular lenses. By consolidating the most current research, this review aims to serve as a resource for clinicians, researchers, and material scientists engaged in the rapidly evolving field of ophthalmology.

## 1. Introduction

The landscape of ophthalmic care has experienced transformative changes over the past few decades, notably due to technological advancements and innovative materials science. Among the burgeoning frontiers in this interdisciplinary field is the application of nanoporous materials, characterized by their unique physicochemical properties and potential for sophisticated functionality [1].

To present an up-to-date overview of this dynamic field, we conducted a comprehensive literature review, explicitly focusing on studies published within the last five years. This review aims to explore the myriad applications of nanoporous materials in ophthalmology, particularly in drug delivery systems (DDS), contact lenses, and intraocular lenses.

Nanoporous materials are distinguished by their high surface area, tunable porosity, and capacity for functional modifications, which make them apt for potentially enhancing the bioavailability and controlled release of therapeutic agents [2,3]. Furthermore, these materials offer some advantages in potentially improving the performance and user experience in ocular devices like contact and intraocular lenses [4,5]. The prospects of these applications, however, are not without challenges; issues related to biocompatibility, manufacturing scalability, and long-term stability warrant thorough investigation [6].

The ensuing sections of this review are structured as follows: Section 2 provides an overview of nanoporous materials, delineating their definitions, characteristics, and unique properties. Section 3 explores the utility of nanoporous materials in the development of drug delivery systems for ocular diseases, examining their role, potential, and types such as nanoporous hydrogels, mesoporous silica, nanoporous gold (NPG), and nanoporous polymers. This section also delves into disease-specific applications and future directions. Section 4 provides an overview of nanoporous materials in intraocular lenses, emphasizing their contributions to biocompatibility and their prospects and roles in cataract surgery. Section 5 focuses on the integration of nanoporous materials in contact lenses, discussing the role of nanoporosity and its impact on user benefits including drug delivery potential. 

By consolidating the most current research and identifying future research trajectories, this review aims to serve as a resource for clinicians, researchers, and material scientists actively engaged in the evolving field of ophthalmology. The synthesis of past, present, and future perspectives on the application of nanoporous materials in ophthalmology offers a roadmap to navigate challenges and harness the potential of these materials in ocular therapeutics.

## 2. Understanding Nanoporous Materials 

### 2.1. Definition and Characteristics

Over the last few decades, research centered around nanomaterials has undergone significant evolution, transitioning from nanomaterial discovery to synthesis and application. The unique properties of these materials, such as their tunable size, versatility, and shape, continue to capture the attention of the scientific community, inspiring further research focused on developing approaches for studying and tuning these structures, particularly in the field of drug delivery [7,8].

Nanoporous materials are a type of nanoscopic material that consist of a well-organized arrangement of nanoscopic pores with diameters of 100 nm or less [9]. Unlike other materials and their nonporous counterparts, they take on unique physical and chemical properties governed by their porosity, size, and surface area, which allow them to tailor their interactions within their environment [10]. Their porous holes, in particular, allow them to encapsulate, store, protect, and deliver large amounts of insoluble drugs to tissues of interest, one of the greatest challenges in ocular drug delivery [11]. According to the International Union of Pure and Applied Chemistry (IUPAC) classification, porous materials are separated into three main categories based on pore size: megaporous materials (>50 nm), mesoporous materials (2–50 nm), and microporous materials (<2 nm) [12,13,14,15], where the term “nanoporous materials” often refers to both mesoporous and microporous materials. Nanoporous materials can take on organic or inorganic frameworks depending on their elemental makeup. Organic nanoporous materials are made from elements, such as boron, carbon, nitrogen, and oxygen [16]. In comparison, inorganic nanoporous materials are made from non-organic and pure metal-type materials, such as zeolites, silicates, ceramics, aluminum, and titanium [17]. While a large amount of inorganic nanoporous materials find utility across different fields, only a few viable organic nanoporous materials can be used mainly due to their lack of biological and chemical stability [18].

### 2.2. The Potential as Ophthalmic Drug Carriers

Some nanoporous materials offer several advantages for addressing current challenges in ophthalmology since their structure can be personalized and controlled for effective drug release [6]. They serve an important role in drug delivery by acting as a therapeutic agent that can encapsulate, hold, protect, and release drugs due to their porous structure [2]. This structural feature allows them to hold a variety of molecules ranging from small biomolecules, such as lipids, to large biomolecules like proteins, and allows them to even protect certain drugs from degradation [19]. 

Nanoporous materials have many unique physiochemical and biological characteristics that make them ideal drug carriers in the delicate ocular environment. First, they are non-toxic and biodegradable and will undergo either enzymatic or non-enzymatic degradation in the body, producing a harmless, biocompartible by-product [20]. This property helps to reduce any potential side effects of a given drug not only during but also after drug delivery. Second, nanoporous materials have an ordered network of pores that vary in size, shape, volume, distribution, and organization, which can all be modified, allowing for fine control over drug load and release [21]. Treatments that require routine drug administration or patient compliance, such as those involving eye drops, can be optimized to achieve the best health outcome using nanoporous materials. Depending on the pH, temperature, enzymatic activity, or other biochemical characteristics of the environment, these pores can be precisely tailored and tuned for capturing and releasing certain drugs [3]. Lastly, they possess a high surface area-to-volume ratio, which allows for high drug absorption and drug bioavailability [3]. With these characteristics, porous materials are able to effectively capture and deliver a large amount of drugs regardless of their solubilities and sizes, a key challenge in the field of drug delivery. Nanoporous materials offer a viable solution for delivering drugs to challenging environments that would typically render the drug ineffective, e.g., the delivery of an insoluble drug to an aqueous environment. They offer promising prospects as an effective, sustainable solution capable of minimizing unwanted side effects, while at the same time improving therapeutic and patient outcomes.

## 3. Challenges with Current Drug Delivery Systems 

Controlled drug delivery systems (DDS) for diseases have been studied across specialties in recent years to overcome the challenges and barriers of traditional drug administrations [22]. While conventional drug administration approaches, such as oral ingestion, injections, or topical application, have been effective thus far, they nevertheless come with drawbacks that limit their full therapeutic effect and outcome [23]. The delivery of ophthalmic drugs to target ocular tissues presents as a unique, complex challenge due to various anatomical barriers. Though multiple administration routes exist, including topical, subconjunctival, periocular, intracameral, intravitreal, and systemic, each possess limitations influenced by specific ocular barriers [23]. Figure 1 illustrates the anatomical barriers to ophthalmic administration.

### 3.1. Principal Routes and Bioavailability

Topical, systemic, and intravitreal deliveries are most often employed to reach the posterior eye segment. Among these, topical applications, which may include solutions, suspensions, ointments, gels, or emulsions, are the simplest yet least efficient, achieving only around 5% penetration to internal eye structures [24]. These ocular barriers, such as the tear film, cornea, and blood-ocular barriers, serve protective roles but simultaneously curtail drug bioavailability [25].

The tear film, consisting of lipid, aqueous, and mucin layers, represents the first obstacle in topical drug delivery (Figure 2). Lacrimal fluid secretion and reflex blinking cause rapid tear turnover, removing a substantial fraction of the topically applied drug. Specifically, the lacrimal fluid turnover rate of approximately 1 µL/min poses a significant limitation on drug retention [24,26].

Nearly 95% of a topically applied drug is eliminated via the nasolacrimal drainage system, comprising the lacrimal sac, canaliculi, and nasolacrimal ducts. This not only reduces ocular availability but also contributes to unwanted systemic absorption [27]. Various factors, such as drug volume and patient age, can influence this absorption rate, necessitating drug delivery designs that optimize ocular surface retention [24].

The cornea’s multi-layered structure (epithelium, Bowman’s membrane, stroma, Descemet’s membrane, and endothelium) acts as both a mechanical and chemical barrier [28] (Figure 3). Its semi-permeable nature allows for passive material transfer, but tight junctions restrict the entry of larger or hydrophilic molecules. The stroma, notably hydrophilic, presents an additional obstacle to lipophilic molecules [24,25].

Although intravitreal administration offers a more direct route to the retina and vitreous, larger and positively charged molecules may find it difficult to cross the retinal pigment epithelium (RPE) barrier [25].

Drug clearance from the aqueous humor occurs via two primary pathways: aqueous turnover through the chamber angle and Schlemm’s canal, and venous outflow from the anterior uvea [29]. While the former is independent of drug properties, the latter is influenced by the drug’s lipophilicity [23].

Both the blood-aqueous barrier (BAB) and the blood-retinal barrier (BRB) serve as major roadblocks for drug delivery to the anterior and posterior segment of the eye [30]. The BAB, associated with the anterior chamber, features endothelial cells and tight junctions that limit drug entry [24]. Conversely, the BRB consists of retinal capillaries and RPEs and presents additional complexity due to the varying permeability of its components [31]. 

### 3.2. Challenges in Delivering Treatments to Ocular Tissues

Bioavailability remains a significant hurdle in delivering medications to the different segments of the eye via systemic, local or topical routes (Figure 4). Systemic medications, whether oral or intravenous, are largely impeded by the blood-retinal barrier (BRB), necessitating high-dose regimens that increase the risk of systemic side effects [32]. Topical formulations and subconjunctival injection, although less invasive, face multiple anatomical barriers that restrict their effectiveness in targeting the posterior eye segment, as elaborated in an earlier section. Intravitreal injections offer a more direct approach but come with their own set of challenges. These injections are invasive and carry risks of severe ocular complications such as endophthalmitis and retinal detachment. However, these risks are minimal when strict protocols and proper techniques are followed. Moreover, their short retention time demands frequent, sterile administrations by an ophthalmologist, thereby reducing patient compliance and elevating healthcare costs [32]. Subretinal injections offer a targeted approach for posterior segment diseases but necessitate vitrectomy surgery under general anesthesia, adding to their invasive nature. Conversely, suprachoroidal injections have garnered interest due to their potential for sustained release and compartmentalization, enabling more precise targeting of specific ocular tissues in the posterior segment. Despite their potential, most applications for suprachoroidal injections remain in early preclinical stages, with their clinical uses currently limited to treating macular edema secondary to uveitis or other ocular pathologies [33].

In summary, the anatomical constraints within the ocular environment necessitate meticulous design and consideration in ophthalmic drug delivery systems. Understanding these barriers is crucial for optimizing bioavailability and therapeutic efficacy. While the current drug delivery methods have their own set of advantages and limitations, the need for a more effective and patient-compliant approach remains an additional critical focus in ophthalmic drug delivery research.

## 4. Nanoporous Materials in Ocular Drug Delivery 

Much attention in the past has been centered around nanoparticle synthesis and design. While still limited, in recent years, researchers have centered their studies around the applications of nanoporous materials as DDS for the treatment of various ocular diseases, such as glaucoma, cataracts, or dry eye [34,35,36]. Common porous carriers, such as nanoporous hydrogel, mesoporous silica, nanoporous silica, and nanoporous nanofibers, have been developed recently and used for the treatment of anterior and posterior segment diseases (Table 1). 

### 4.1. Organic Nanoporous Materials for Ocular Drug Delivery

#### Nanoporous Hydrogels for Ocular Drug Delivery

Hydrogels consist of three-dimension networks of polymer chains that can absorb a large amount of water, at least 10% of their total weight or volume, while maintaining their structural integrity [44]. Their flexibility is similar to that of natural human tissue, and they can be chemically modified based on the characteristics of their environments, such as temperature, pH, solvent composition, and electrical field presence [45,46]. Due to their high porosity, which can be adjusted through crosslinking or by modifying their affinity for the aqueous environment, hydrogels serve as excellent DDS that are highly permeable to different types of molecules and drugs [47]. For instance, although the most common treatment approach for cornea abrasion (CA) includes antibiotic drops and ointments, they often do not work as anticipated or as effectively, due to the lack of patient compliance [48]. Luo et al. (2022) [37] therefore developed a therapeutic hydrogel sheet (THS) composed of a functional hydrogel, consisting of a poly(hydroxyethyl methacrylate), positively-charged chitosan, and zinc oxide nanoparticles, and a ternary drug-carrier system, consisting of dipalmitoylphosphatidylcholine liposome (DPPC), nanoparticles with epigallocatechin gallate (EGCG), and hyaluronic acid nanoparticles to improve treatment for CA. After tailoring the degradability, the researcher found that the THS could be specifically tailored for multistage drug release. Their results showed that the THS decreased the inflammatory response at the beginning stage, promoted wound healing at the middle stage, and prevented scar formation at the final stage that follows CA. They validated this model in rabbits and showed that the percent recovery was 90%, which represents more than an eight-fold increase compared to that of conventional eye drops. Although recent literature remains limited, the application of nanoporous hydrogels holds considerable potential as a major component in ocular DDS, offering avenues for improving treatment precision and clinical outcomes. To build off the current study, research efforts should be centered around investigating if the DDS developed by Luo et al. (2022) is compatible with other ocular drugs and could be used for treating more complex ocular diseases.

### 4.2. Inorganic Nanoparticles for Ocular Drug Delivery

#### 4.2.1. Mesoporous Silica for Ocular Drug Delivery

Mesoporous silica and nanoporous silica are inorganic nanoporous materials, synthesized from sodium silicates or silica tetraethyl orthosilicate with a surfactant micelle [49]. They have pore sizes ranging from 2 nm to 50 nm and <2 nm in diameter, respectively, which can be tuned based on various environmental influences, such as the morphology of surfactants, temperature, and pH conditions [50]. Their high surface area, uniform yet tunable pore size, biocompatibility, pore volume, and convenient manufacturing process present them as suitable carriers for ocular drug delivery [51,52,53]. 

In light of these developments, recent studies have investigated innovative drug delivery systems such as hollow mesoporous organosilica (HOS) nanocapsules for co-delivering NO donor drugs to the eye as potential glaucoma treatments. Glaucoma is a complex, multifactorial ocular disease often characterized by the progressive degeneration of the optic nerve. Although various risk factors have been implicated, intraocular pressure (IOP) remains the most important modifiable risk factor. Traditional medical and surgical interventions predominantly focus on lowering IOP to halt or slow the disease’s progression. As our understanding of glaucoma deepens, emerging research is exploring the role of physiological mediators like nitric oxide (NO) in disease management. NO is not only involved in reducing IOP but may also have a role in improving the perfusion of the optic nerve head, offering a multi-faceted approach to treatment. Latanoprostene bunod 0.024%, for instance, is a therapeutic agent that has demonstrated promise in clinical trials in this regard [54]. In one study, Fan et al. (2021) [34] investigated the use of hollow mesoporous organosilica (HOS) nanocapsules as a DDS for co-delivering nitrogen oxide (NO) donor drugs hydrophobic JS-K (diazeniumdiolates) (J_R_) and hydrophilic L-Arginine (L_o_) to the eye as a potential treatment for glaucoma. They loaded J_R_ and L_o_ into the internal cavity and mesoporous shell and found that the synthesized silica showed high stability and biocompatibility during degradation. After testing the HOS’s ability to deliver NO donors (J_R_/L_O_) to the target tissue on a Cav1 knockout (KO) mice, which was observed to have higher intraocular pressure (IOP) compared to a Cav1 mice, they found that the HOS showed greater tissue permeability and was able to directly transport the drug to the tissue of interest. Additionally, Fan and collaborators investigated the long-term side effects of HOS-based NO nanotherapeutics as current sustained NO donor treatments tend to release NO in excess, which can lead to cell damage. After treating the Cav1 KO mice with the HOS-J_R_L_O_, every 48 h, they found that across 10 days, the IOP still decreased, showing that the HOS was able to release the NO in a sustained and stable fashion, thus preventing tissue damage, while still decreasing IOP [55]. They found similar results when validating the generality of the HOS-based NO nanotherapeutics on another ocular hypertension mice model. Subsequent analyses showed that the HOS-based NO nanotherapeutics model regulated the IOP reduction through different methods, such as decreasing humor outflow resistance. It would be critical to know the exact factors in different models that affect IOP reduction. Future research should focus on validating this model using other medications and in other disease mice models. 

In another study, Sun et al. (2019) [38] looked at the potential of using mesoporous silica nanoparticles as a nanodrug delivery system of bevacizumab to improve antiangiogenic therapy for retinal and choroidal neovascularisation. This pathological condition is associated with ophthalmic diseases such as ischemic retinal vein occlusion (RVO), proliferative diabetic retinopathy (PDR), and wet age-related macular degeneration (wet-ARMD). These conditions can lead to severe consequences like intravitreal hemorrhage, retinal detachment, and neovascular glaucoma, all of which pose acute risks of blindness. Bevacizumab, one among several anti-VEGF treatments that include agents such as aflibercept, ranibizumab, and faricimab, possesses antiangiogenic properties. These agents have collectively demonstrated efficacy in reducing neovascularization and inducing its regression. Presently, the standard approach for administering treatments like bevacizumab is through intravitreal injections. However, this method necessitates frequent visits to ophthalmologists—often on a monthly basis—due to the drug’s short half-life and lack of sustained release when injected into the vitreous. Various studies have thus concentrated on methods for achieving a more sustained release of bevacizumab, exploring alternative administrative routes such as suprachoroidal injection techniques [33], or employing nanobased drug delivery systems [1,5]. In their study, Sun et al. prepared mesoporous silica nanoparticles loaded with bevacizumab and reported encapsulation and drug loading efficiencies of 70.4% and 79.2%, respectively [38]. They found that MSN encapsulation was able to effectively preserve the bevacizumab in the vitreous/aqueous humor, maintaining its drug concentration; avoid tissue toxicity; suppress vascular endothelial growth factor-induced endothelial cell proliferation, migration, and tube formation in vitro; and provide sustained inhabitation of corneal neovascularization and retinal neovascularization in vivo. As MSN drug encapsulation presents as a promising DDS for intraocular neovascular diseases, future research should focus on testing this approach using different antiangiogenic therapy agents.

In another study, Pavia et al. (2021) [56] also looked at the application of mesoporous silica nanoparticle encapsulation. They incorporated tacrolimus (TAC) into silica nanoparticles functionalized with 3-aminopropyltriethoxysilane (MSNAPTES) as a potential treatment for general posterior and anterior segment eye diseases They investigated the applicability and safety of the systems through in vitro and in vivo studies, reporting that the MSNAPTES achieved up to 7% TAC loading. After looking at the vitreous cavity injection of the MSNAPTES over 15 days, no retinal impairments or optical nerve atrophy were observed, thus confirming the application of MSNAPTES as a promising and effective carrier for TAC for the treatment of ocular diseases.

Lastly, Wu et al. (2020) [40] were the first at the time to look into using mesoporous silica nanoparticles for drug delivery of mitomycin C to prevent postoperative pterygium recurrence. Wu and collaborators coupled low-density lipoprotein (LDL) with mesoparticles loaded with mitomycin C (MMC) (MMC@MSNs-LDL) for the inhibition of pterygium subconjunctival fibroblasts. They found the MMC loading efficiency to be 6% and that the MMC@MSNs-LDL had effective MMC targeting, thus inhibiting abnormal proliferation. Wu and collaborators reported that the MMC@MSNs-LDL exhibited less toxicity compared to normal fibroblasts, further emphasizing it as a precise DDS. Future studies should focus on evaluating the long-term effect and safety of this drug carrier in animal models.

#### 4.2.2. Nanoporous Polymers for Ocular Drug Delivery

Nanoporous polymers are different types of polymers, such as nylon, polyethylene, and polyester, that consist of a framework of pores with sizes below 200 nm [57]. They take on unique physical and chemical properties unlike other materials, such as thermal stability, hardness, conductivity, and a large surface area which helps enhance their ability to absorb molecules, such as drugs. Due to their low cost, biocompatibility, and biodegradable properties, these materials have been especially studied in polymeric DDS [58,59,60,61]. 

In one study, He et al. (2021) [41] developed a dosage-controllable drug delivery implant, designed for placement within the vitreous cavity. This implant consisted of a nanoporous PLGA capsule and light-activated liposomes, specifically for delivering methotrexate (MTX). Using pulsed near-infrared (NIR) laser both in vitro and in vivo, they were able to achieve controllable MTX releases, with release patterns following zero-order kinetics for the 1000 μg dose and first-order kinetics for the 500 μg dose. They observed that the drug releasing behavior was consistent between the in vitro and in vivo settings. Importantly, histological analyses revealed no abnormalities in the eyes or evidence of cytotoxicity, immune responses, or foreign body reactions after implantation into the vitreous cavity and treatment, thereby attesting to the safety of this ocular delivery system.

#### 4.2.3. Nanoporous Nanofibers for Ocular Drug Delivery

Nanoporous nanofibers are materials with a nanoscale structure and porous architecture. They are produced through electrospinning, a process where a polymer solution is exposed to an electric field, which causes the fibers to stretch and form their unique fibrous structure [62]. The porous architecture which also results from this fabrication approach can take on a variety of forms, ranging from close-pore structures to open-pore structures [63]. 

While research using nanoporous nanofibers as DDS is still very new and limited, in one recent study, Rohde et al. (2022) [42] investigated the potential applications of nanofibrous electrospun scaffolds, consisting of polymers, as a new DDS system for overcoming the limitations of liquid and semi-solid formulas, which are the most common drug delivery systems of ophthalmic diseases. They investigated this system specifically as a drug release system delivering gentamicin and dexamethasone for bacterial conjunctivitis. They found that when the nanofibers came in contact with the ocular surface, they dissolved the tear fluid. They also found that compared to the current models used for this type of drug delivery, the new system showed higher dosing accuracy and drug recovery. Subsequent analyses showed that gentamicin released from the fibers inhibited the growth of disease-specific bacteria and showed full antibacterial activity over a 12-week shelf-life period. After assessing the system in a porcine ex vivo microfluidic cornea model, they found a significantly prolonged ocular residence time compared to a conventional eye drop. 

In another study, Li et al. (2023) [43] designed and developed a class of nanoporous, nanofibrous membranes loaded with celastrol for a sustained release and with hyaluronic acid to prevent burst release using electrostatic spinning. After placing the film directly in the subconjunctiva after injury, the membranes appeared to inhibit subconjunctival fibrosis. As celastrol was recently found to induce autophagy [64], a promising novel therapeutic target in various diseases, the researchers further hypothesized that celastrol could be a therapeutic option for the long-term development of antifibrotic drugs. These antifibrotic drugs may potentially help prevent tissue fibrosis following filtrative glaucoma surgery (e.g., trabeculectomy), a procedure aimed at reducing intraocular pressure (IOP) in glaucoma patients. Further analyses showed that celastro indeed triggered autophagy and promoted the expression of LC3A, LC3B, and Beclin-1 and that celastrol induced autophagy by inhibiting the PI3K/Akt/mTOR kinase complexes. After finding that celastrol decreases the expression and signaling of TGF-beta1, which is part of the signaling complex TGF-beta1/Smad2/3, a key role in fibrosis, the researchers concluded that celastrol inhibits subconjunctival fibrosis by inducing autophagy through inhibiting the PI3K/Akt/mTOR signaling and potentially also the TGF-beta1/Smad2/3 signaling pathway. 

While nanoporous nanofibers find many of the same applications as other porous structures due to their unique porous structure, such as filtration, catalysis, sensors, energy storage, etc., they also have the additional advantageous properties of nanofibers, such as flexibility and a lightweight build, which make their use in ocular DDS very effective [65].

Advancements in nanotechnology within the past decades have made great strides in improving treatments for ocular diseases, from improving drug bioavailability to decreasing eye irritation and improving ocular biocompatibility [66,67,68,69]. As mentioned in the previous section, nanocarriers present as promising tools for drug delivery due to their uniform distribution, stability, high drug load capacity, enhanced biocompatibility, biodegradability, controlled release, increased cellular update, manufacturing ease, and cost effectiveness [1]. Nanoporous materials, in particular, further enhance these characteristics of nanocarriers, since their porous structure allows for even greater drug loading capacity, more precise control of drug release over extended periods, and protection of the encapsulated drugs from degradation [2].

## 5. Nanoporous Materials in Intraocular Lenses 

Although still in its infancy, the research for the use of nanoporous materials in intraocular lenses (IOL) shows significant promise for applications such as drug depots, biomarker sensors, lens structure enhancement, and the prevention of posterior capsular opacification (PCO).

### 5.1. Structural and Functional Benefits of Nanoporous Intraocular Lenses

Liang et al. have made structural improvements to an accommodating IOL by adding a porous support ring made of PDMS (polydimethylsiloxane) between the front and back support meniscus. This modification improves lens stability as less central liquid polymer is needed. Given that PDMS is a common IOL material, toxicity is not a concern; however, the clinical use for this novel design has not yet been established as optical quality testing has not been conducted [70]. 

Nanoporosity can play a role in the detection of disease biomarkers in the aqueous humor (AQ). A multifunctional nanoporous (NP) IOL composed of a PEGDAAm (diacrylamide-group-modified poly(ethylene glycol)) hydrogel was developed by Shin et al. with a mechanism to detect MMP-9 (matrix metalloproteinase-9). The IOL was embedded within an MMP-9 activatable fluorogenic sensor. Fluorescence detected after intraocular injection of MMP-9 demonstrated the efficacy of this novel technology. Visual field distortion could potentially be a limiting factor for clinical use as the effects of the fluorescence on optical quality were not measured [71]. Kim et al. developed an NP IOL with a moiré pattern biomarker sensor. An acrylamide, acrylic acid, and methylenebisacrylamide hydrogel with a moiré pattern was embedded with anti-BDNF (brain-derived neurotrophic factor) antibodies. When the BDNF protein passes from the AQ through the IOL pores and binds to the antibody, hydrogel crosslinking occurs, shrinking and altering the moiré pattern. These changes can then be detected through microscopy. No optical quality testing was conducted [72], but this mechanism is likely not as visually intrusive as intraocular fluorescence. PCO is a common post-cataract surgery complication developing from the migration of lens epithelial cells (LECs) from the periphery to the visual axis of the lens posterior capsule [73]. An IOL with a surrounding nanoporous gold (NPG) ring was developed by Liu et al. for the prevention of PCO. Irradiation of the NPG eliminates the LEC cells before migration, reducing PCO development. Although no corneal damage was noted in a rabbit model [4], the risk of ocular and visual damage may still exist from the laser irradiation. Furthermore, bacterial biofilm formation tends to occur toward the periphery of IOL surfaces [74], and given that porous material allows better bacterial seeding [75], this may pose challenges with infection when the NPG is not subject to irradiation.

Please refer to Table 2, “Nanoporous Intraocular Lenses (IOL): Structural and Functional Benefits”, for a detailed analysis of the enhanced structural and functional attributes of nanoporous IOLs.

### 5.2. IOL Drug Delivery Potential

The self-administration of eye drops poses a challenge for patients, including the need for frequent administration, significant medication loss through spillage and leakage, and systemic absorption through the nasolacrimal duct. Hand-eye discoordination poses issues as well. The lack of medication adherence is linked with poor ocular health outcomes [76].

#### Drug Delivery Studies (Table 3)

A mechanism for controlled and non-invasive drug release to the eye can help improve ocular health outcomes. Implantable ocular drug depot devices delivered through intravitreal injection currently exist on the market [77]; however, the risk of severe complications including endophthalmitis, although rare, is possible [78]. Preliminary research on NP drug-eluting IOLs shows promise as an alternate method for drug delivery. Karamitsos et al. developed a NP, single-layer film embedded with dexamethasone placed over an acrylate/methacrylate and PMMA (poly(methyl methacrylate)) polymer lens component. An eight-week dexamethasone drug release study reveals three separate elution phases with different release kinetics: an initial burst release, an exponential release phase, followed by steady release [35]. This release instability may only be useful with conditions needing an initial drug loading phase. The optical properties of the IOL were largely unaffected.

Please refer to Table 3, “Nanoporous Intraocular Lenses (IOL): Drug Delivery Potential,” for an in-depth exploration of the therapeutic capabilities of nanoporous IOLs in drug administration.

**Table 3 ijms-24-15599-t003:** Nanoporous intraocular lenses (IOL): drug delivery potential.

IOL Material Type	Key Features	Challenges	Current Usage Status	References
Crystalline films of sulfonated syndiotactic polystyrene (s-PS)	- Potential drug release depot (both hydrophilic and hydrophobic).	Theoretical drug depot capacity, not yet tested	Not currently in clinical use, research use only	[79]
poly(2-hydroxyethyl methacrylate) (pHEMA) hydrogel	- Developed a reliable method to imprint pHEMA hydrogel with nanopillar pores and adjust the number, shape, and arrangement	Potential for porous pHEMA to calcify over long periods of time [13]	pHEMA hydrogel commonly used in IOLs pHEMA with nanopillar pores not currently in clinical use, research use only	[80]
Acrylate/methacrylate copolymer with Poly(methyl methacrylate) (PMMA) haptic part and nanoporous dexamethasone film	IOL optical properties largely unaffected by dexamethasone embedded film	Overall unstable drug release: three separate drug elution phases (burst, exponential, and steady release)	Not currently in clinical use, research use only	[35]

### 5.3. Potential Drug Delivery NP Materials

Although limited, a couple recent studies have centered around NP IOL development techniques. Both need further testing to determine optical quality, drug capacity, and drug release kinetics. Krajňák et al. developed an efficient mechanism to build NP IOLs in a pHEMA (poly(2-hydroxyethyl methacrylate)) hydrogel using a replica mold. The nanopores, coined “nanopillars”, are adjustable in number, shape, and arrangement, potentially allowing for a drug storage depot adjustable based on ocular drug needs. Given that pHEMA has a long history of use in ophthalmic applications, toxicity is not expected, and the study’s preliminary in vitro cytotoxicity testing with fibroblasts corroborates this [80]. The challenge with pHEMA is the potential for calcification over time [81] and porosity may seed additional calcification. IOLs developed from an s-PS (syndiotactic polystyrene) polymer material by Zuppolini et al. were flexible, hydrophilic, and durable. The flexibility of s-PS makes it a potential IOL for use in micro-incision cataract surgery, while its NPs and hydrophilicity gives it potential as a drug depot. Sulfonation of polystyrene leads to less cellular adhesion using an NIH3T3 cell line compared to a non-sulfonated polystyrene polymer, potentially reducing the risk of PCO. These findings may not be clinically significant as cell adhesion to common IOL materials like silicone and PMMA was not measured nor compared to s-PS [79]. Thiolated cyclodextrins (t-CDs) developed within a NP IOL could also serve as an effective drug delivery system. No IOLs have been developed with t-CDs to date, but Kesavan et al. have amalgamated dexamethasone embedded cyclodextrins (CDs) within a Carbopol 980 NF and sodium carboxymethylcellulose hydrogel with promising therapeutic results in a rabbit model. As CDs improve solubility of hydrophobic drugs and t-CDs can improve the residence time of drugs on the ocular surface through mucoadhesive properties, this technology could potentially enhance ocular drug delivery from an IOL or Cl drug depot.

### 5.4. Challenges and Future Considerations

Given biocompatibility is an important consideration when developing ocular devices, the need for additional safety and efficacy testing in animal and human models for these novel IOLs remains. As porous ocular implants allow for easier bacterial adhesion [75], future research should explore the risk for bacterial adhesion to the NP IOL. Furthermore, replenishing a depleted drug depot remains an issue. Without a non-surgical method to replete the IOL drug depot, the only benefit is using these devices in ocular disease only requiring a short course of medication therapy, up to a maximum of 1–2 months depending on the depot capacity. This could fill a short-term need for ocular antibiotics or anti-inflammatory drugs.

## 6. Nanoporous Materials in Contact Lenses 

Eye drops have poor ocular penetration and high systemic absorption, while hand-eye discoordination can make administration difficult. Improved drug delivery systems into the eye can improve therapeutic outcomes through better adherence. Like NP IOLs, NP contact lenses (CL) can address some of the challenges faced with topical ocular drug administration. CL-mediated drug delivery allows longer drug residence time on the cornea, improving penetration through ocular barriers while reducing the likelihood of leakage out of the eye through lacrimal fluid secretion and nasolacrimal drainage. Bimatoprost [82] and latanoprost [83] eluting CLs are examples currently in clinical trials. This CL drug delivery via embedded liposomes, polymeric nanoparticles, and cyclodextrins [84] has limits on total drug capacity simply due to physical limitations in surface area. Here, we explore the functionality that nanoporosity can add to CLs.

### 6.1. Contact Lens Drug Delivery Potential

NP contact lenses offer a less invasive and more practical option for drug delivery compared to NP IOLs. A 2-HEMA (2-hydroxylethylmethacrylate) CL with cyclosporine A (CsA) embedded NP silica may reduce the irritation commonly experienced with CsA eye drops. This CL developed by Choi et al. shows CsA elution reaches the cornea and conjunctiva. For dry eye disease, this is not a significant finding given the drug does not need to pass through the corneal barrier for therapeutic effect. Sustained release of CsA only lasts 2 days, a disadvantage of this design [85]. An NP, self-adaptive drug delivery system was developed by Ding et al. by embedding PDMS (polydimethylsiloxane) microtubes in a PDMS CL hydrogel. As intraocular pressure (IOP) increases, the contact lens stretches, enlarging the microtubes and increasing timolol release. After an initial burst release of timolol, up to 40 days of steady drug release occurs. Steady and controlled drug release may minimize systemic absorption, an advantage for medications like beta blockers where systemic absorption leads to bradycardia and low blood pressure. The ideal drug size and solubility characteristics were not studied [86]. An alternative to a fully NP CL is a thin nanoporous films that can be placed over a CL. Yin et al. developed a porous BSA (bovine serum albumin) fibrous film to evaluate its drug delivery capacity for potential future use in CLs. The BSA embedded with kaempferol demonstrates in vivo anti-inflammatory effects but given the short duration of release (>340 h) and poor optical transparency, this material is not ideal for future testing [87]. 

In a recent study, Lai et al. (2023) [36] investigated pH-trigged drug-eluting contact lenses (DCLs) in combination with mesoporous silica nanoparticles (LPMSNs) as a potential therapeutical DDS for glaucoma. They specifically studied the loading and releasing behavior of two specific glaucoma drugs, timolol maleate salt (TMS) and brimonidine tartrate salt (BTS). They found that the LPMSN-laden DCLs were able to prevent drug loss during sterilization and storage and did not require separate drug loading, showing compatibility with the traditional manufacturing process of CLs. The LMSNs exhibited large surface (660 m^2^ g^−1^) and high pore volume (1.41 cm^3^ g^−1^), which allowed for a substantial loading capacity for and release of the glaucoma drugs, TMS and BTS. After measuring the drug release behavior of the DCLs in greater detail, they found that the LPMSN-laden DCLs released the TMS and BTS at slower rates in comparison to the standard CLs, which allowed for longer drug release time (up to multiple days) and greater ocular availability. It would be important in future research to build upon this study and look at specific lenses that could be worn for multiple days at a time that would allow for this sustained drug release since this was beyond the scope of the study. Further analyses revealed that the LPMSN-laden DCLs exhibited good biocompatibility and non-toxicology, thus supporting their potential use in clinical applications. 

In another recent study, Lai et al. (2023) [88] created hydrogel contact lenses (CLs) using large-pore mesoporous silica nanoparticles (LPMSNs) functionalized with amine groups (LPMSN-amine) for the delivery of hyaluronic acid (HA) in the treatment of dry eye syndrome. Conventional first-line treatments for dry eye syndrome typically involve the use of artificial tears. However, these treatments require frequent re-application, and their efficacy is contingent upon patient compliance. A sustained-release approach could mitigate these limitations by potentially providing continuous relief without requiring frequent application. In an effort to overcome the limitations of conventional mesoporous silica nanoparticles, which have pore sizes too small to carry high molecular weight molecules like HA, Lai et al. utilized LPMSNs with pores larger than 30 nm. By functionalizing these LPMSNs with amine groups, they achieved a 12.6-fold increase in HA loading and superior release capabilities compared to standard CLs. This was most likely explained by the strong attraction between the positively charged LPMSN-amine and the negatively charged HA. Not only were these characteristics improved, but the optical and physical properties of the CLs were not compromised. The lenses also maintained moisture, a critical characteristic for CLs to treat dry eye. These LPMSN-amine contact lenses serve as good candidates for large-molecule ocular drug delivery, especially for treating dry eye syndrome and providing comfort to patients.

Please refer to Table 4, “Nanoporous Contact Lenses (CL): Drug Delivery Potential”, for an overall assessment of the drug delivery capabilities of nanoporous contact lenses.

### 6.2. Development of Nanoporous CL Materials 

Oh et al. developed a reliable manufacturing approach for a HEMA (2-hydroxyethyl methacrylate) and MAA (methacrylic acid) porous contact lens using various blowout agents [89]. Further testing should be conducted to determine the potential for ocular irritation, NP drug depot capacity, and drug release kinetics.

### 6.3. Functional Benefits of Nanoporous Contact Lenses

Nanoporosity can provide additional structural and functional benefits to CLs. Song, Ben-Shlomo, and Que developed a tri-functional PDMS (polydimethylsiloxane) based CL enveloped with a porous anodic aluminum oxide (AAO) thin film. This CL can measure IOP through optical signal shift. As intraocular pressure changes, corneal and contact lens curvature shifts, and this change can be measured and translated to an IOP measurement. The function holds no clinical significance as It must be performed ex vivo. The CL nanopores can also function as drug depots, providing stable drug release with 90% of the depot capacity released after 30 days. Lastly, this CL can detect biomarkers. The binding of a target biomarker to an antibody embedded in the CL can be detected through the change in an optical reflection signal before and after binding [90]. In 2020, Lee et al. developed a less invasive IOP measurement technique with use of a thermosensitive, moiré-patterned, drug-eluting nanoporous CL. As IOP changes affect eyeball diameter, the consequent changes in the CL moiré patterns can be photographed and translated to an IOP reading. The nanopores function as drug reservoirs. Drug release occurs after insertion of the CL into the rabbit model, initiated by warmer body temperature. Timolol presence in the rabbit aqueous humor over 7 days with a significant reduction in IOP suggests therapeutic potential. There is uncertainty about whether the lens can continue delivering drug past 7 days as the timolol measurements were discontinued [91].

Please refer to Table 5, “Nanoporous Contact Lenses (CL): Functional Benefits” for an review of the functional advantages offered by nanoporous contact lenses.

### 6.4. Challenges and Future Considerations

The field of nanoporous contact lenses is still underdeveloped and questions about pharmacokinetics, cost, and manufacturing remain unanswered. Furthermore, the use of multiple topical medications is common in some ocular diseases, yet the evaluation of some multi-drug depots has not been conducted. Additional research is needed to better understand the future clinical role of NP CLs.

## 7. Conclusions

As the field of ophthalmology continues to evolve, the integration of nanoporous materials stands as a testament to the transformative power of interdisciplinary research. These materials, distinguished by their high surface area, tunable porosity, and functional adaptability, have opened new horizons in the treatment and management of ocular diseases. Their applications span from innovative drug delivery systems to the enhancement of intraocular and contact lenses, offering the potential for more effective therapies and improved patient experiences.

By consolidating the most recent advances and outlining potential future directions, this review has aimed to provide a comprehensive and current understanding of the role of nanoporous materials in ophthalmology. It serves as a resource and guidepost for clinicians, researchers, and material scientists who are shaping the future of this dynamic and impactful field. The potential of nanoporous materials underscores the need for interdisciplinary collaboration between scientists and clinicians. With a focus on enhancing therapeutic modalities, the ultimate goal is the betterment of patient vision and overall quality of life.

## Figures and Tables

**Figure 1 ijms-24-15599-f001:**
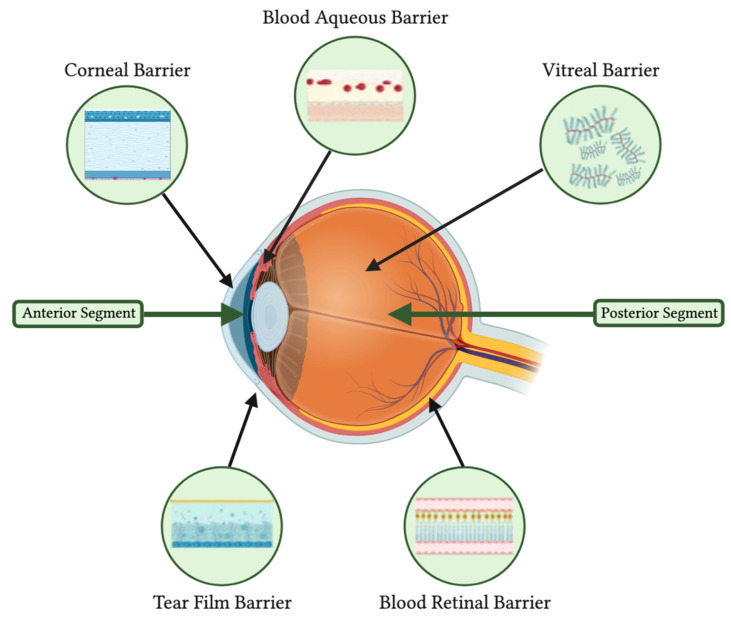
Structural impediments to pharmacologic agents in ophthalmic administration. (Created by Wu et al., via BioRender, https://app.biorender.com/, on 16 February 2023).

**Figure 2 ijms-24-15599-f002:**
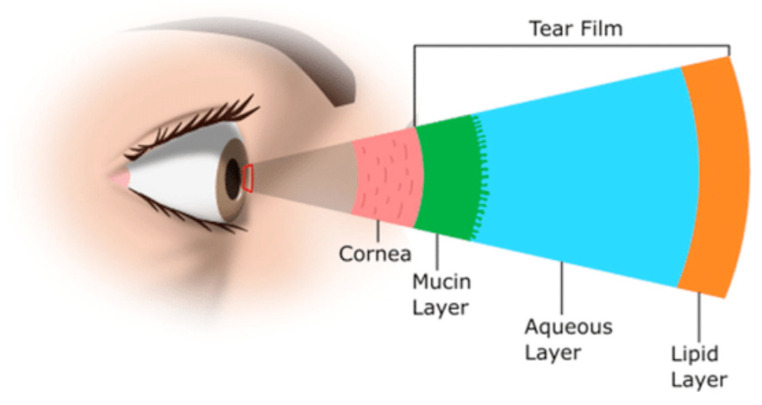
Layers of the tear film as the initial barrier in topical drug delivery. This illustration highlights the lipid, aqueous, and mucin components that form the tear film. (Created by Wu et al., via BioRender, https://app.biorender.com/, on 16 June 2023).

**Figure 3 ijms-24-15599-f003:**
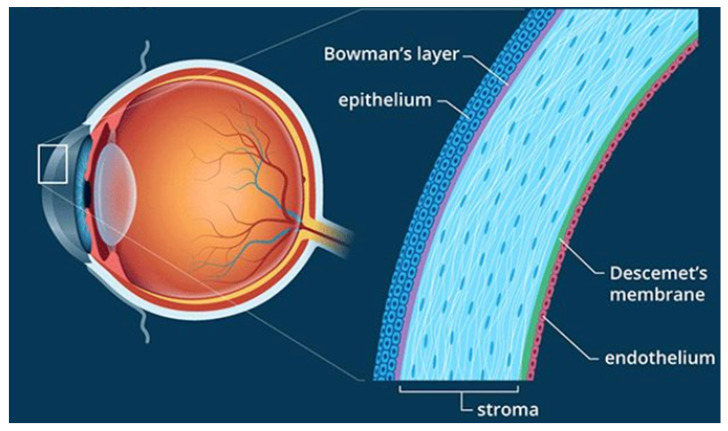
The multi-layered structure of the cornea as a dual mechanical and chemical barrier. Depicted are the epithelium, Bowman’s membrane, stroma, Descemet’s membrane, and endothelium. (Created by Wu et al., via BioRender, https://app.biorender.com/, on 10 June 2023).

**Figure 4 ijms-24-15599-f004:**
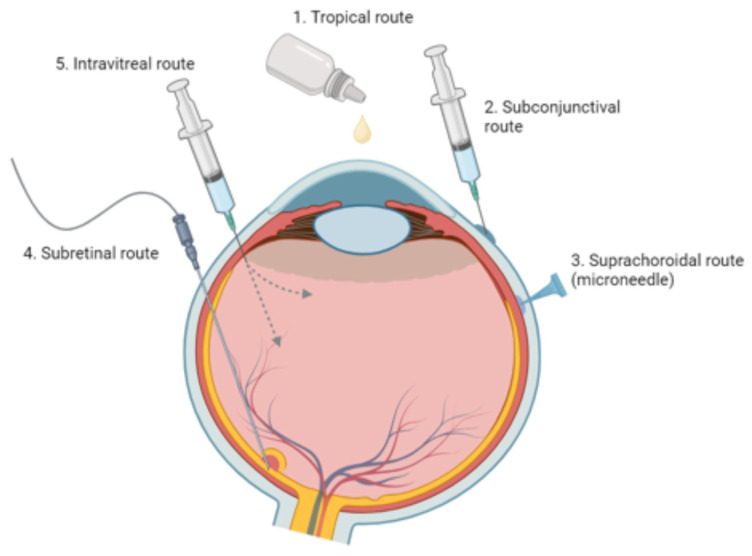
Ophthalmic medication delivery methods. This illustration showcases topical, subconjunctival, intravitreal, suprachoroidal, and subretinal administration techniques. (Created by Wu et al., via BioRender, https://app.biorender.com/, on 10 April 2023).

**Table 1 ijms-24-15599-t001:** Nanoporous DDS for ocular diseases.

Indication	Drug	DDS	Advantages and Considerations	Administration Route	Stage	Reference
Corneal abrasion	Epigallocatechin gallate (EGCG); 8-1,3-glucan; and SB431542	Nanoporous hydrogels	Controlled and sequential release of multiple drugs tailored to address different stages of corneal tissue repair.	Topical	Preclinical—in vitro	[37]
Glaucoma	Nitric oxide (NO)	Mesoporous silica nanoparticles	Increased tissue permeability; tissue targeting; and drug penetration. Sustainable and stable delivery of NO to tissue site.	Topical	Preclinical—in vitro and in vivo	[34]
Ocular neovascularisation	Bevacizumab	Mesoporous silica nanoparticles	Effective preservation of bevacizumab concentration, without causing toxicity to tissues.	Injection (subconjunctival)	Preclinical—in vitro and in vivo	[38]
Posterior uveitis	Tacrolimus	Mesoporous silica nanoparticles	DDS achieved up to 7% TAC loading, without any damage to the retinal tissue or optic nerve tissue.	Injection (intravitreal)	Preclinical—in vitro and in vivo	[39]
Postoperative pterygium recurrence	Mitomycin C (MMC)	Mesoporous silica nanoparticles	Targeting and delivery of MMC was effective and the nanoparticles exhibited less toxicity compared to normal fibroblasts.	Injection (subconjunctival)	Preclinical—in vitro and in vivo	[40]
Posterior uveitis	Methotrexate (MTX)	Nanoporous polymers	DDS did not show any toxicity, immune or foreign body responses following implantation.	Implantation	Preclinical—in vitro and in vivo	[41]
Conjunctivitis	Gentamicin and dexamethasone	Nanoporous Naofibers	Prolonged resistance time, increased tear fluid viscosity and contact time.	Topical	Preclinical—in vitro and ex vivo	[42]
Subconjunctival fibrosis	Celastrol	Nanoporous nanofibers	DDS prevented burse release of celastrol and was still able to preserve the important PI3K/Akt/mTOR pathway-inhibiting effect of celastrol.	Implantation	Preclinical—in vitro and in vivo	[43]

**Table 2 ijms-24-15599-t002:** Nanoporous intraocular lenses (IOL): structural and functional benefits.

IOL Material Type	Key Features	Challenges	Current Usage Status	References
Accommodating IOL: PDMS (Polydimethylsiloxane) polymer layers surrounding porous support ring and optical liquid	- Improved stability for an accommodating lens (less central liquid polymer component required)	Clinical use not yet established	Not currently in clinical use, research use only	[70]
PEGDAAm hydrogel (Diacrylamide group-modified PEG diacrylamide)	- Base prototype for an intraocular lens acting as a sensor for ocular biomarkers	Biomarker diffusion dependent on hydrogel density (nanopore size)	Not currently in clinical use, research use only	[71]
Acrylamide, acrylic acid, & methylenebisacrylamide hydrogel	- Novel mechanism for ocular biomarker detection - Ex vivo potential established - Theoretically limited risk of in vivo harm	Potentially negative effect on visual field (unknown)	Not currently in clinical use, research use only	[72]
Polymethyl methacrylate (PMMA) with nanoporous gold ring (NPG)	- Novel method to reduce posterior capsular opacification (PCO) occurrence	- Rabbit model demonstration of safety only - NPG may increase risk of bacterial biofilm formation [6,7]	Not currently in clinical use, research use only	[4]

**Table 4 ijms-24-15599-t004:** Nanoporous contact lenses (CL): drug delivery potential.

CL Material Type	Key Features	Challenges	Current Usage Status	References
2-Hydroxylethylmethacrylate (2-HEMA) polymer base with nanoporous silica containing cyclosporine A (CsA)	- CsA eluting contact lenses do not appear to cause discomfort in a rabbit model given topical CsA is known to cause ocular irritation	- Sustained release of CsA only occurs for a short period (48 h)	Not currently in clinical use, research use only	[85]
PDMS (Polydimethylsiloxane) lens with embedded PDMS microtubes	- Up to 40 days of drug release (capacity can be altered based on microtube size) - Adaptive drug release function: as IOP increases, contact lens stretches, increasing drug delivery -Steady drug release kinetics after larger initial drug release	- Long-term drug stability in the microtube depot is not considered - Preference for depot to house hydrophilic, hydrophobic, and amphipathic compounds not specified	Not currently in clinical use, research use only	[86]
HEMA (2- hydroxyethyl methacrylate) and MAA (methacrylic acid) lens base	- Demonstrates relationship between choice of blowout agent and resulting hydrogel water content and index of refraction - A reliable approach to manufacturing a porous contact lens	- No in vivo testing for ocular irritation: limited clinical applicability	Not currently in clinical use, research use only	[89]
Not a contact lens. Preliminary analysis of a BSA porous film to investigate its drug delivery capacity (Kaempferol).	- BSA/KAE combination demonstrates in vivo anti-inflammatory and antioxidant effects	- Short duration release profile (>340 h) - Poor film transparency: will likely affect vision quality	Not currently in clinical use, research use only	[87]

**Table 5 ijms-24-15599-t005:** Nanoporous contact lenses (CL): functional benefits.

CL Material Type	Key Features	Challenges	Current Usage Status	References
PDMS (Polydimethylsiloxane) silicone elastomer embedded with anodic aluminum oxide (AAO) thin film	- Nanopores function as pressure sensors for monitoring IOP, drug reservoir, and biomarker detection sensors	- Potential effect on visual field - Likely very expensive if ever commercialized	Not currently in clinical use, research use only	[90]
HEMA (hydroxyethylmethacrylate) with timolol-loaded thermosensitive PNIPAM (poly(N-isopropylacrylamide))	- Multifunctional nanopore: drug reservoir and can measure IOP - Unique temperature activated drug release - Demonstrates drug penetration into aqueous humor with therapeutic IOP reductions	- Potentially limited drug reservoir capacity: release past 7 days unknown	Not currently in clinical use, research use only	[91]
Timolol maleate salt (TMS) and brimonidine tartrate salt (BTS) loaded in mesoporous silica nanoparticles	- Prolonged resistance time - No preloading required - Improved drug load efficiency - Demonstrated biocompatibility	- It is unknown if these specific lenses could be worn for multiple days at a time that would allow for sustained drug release	Not currently in clinical use, research use only	[88]
Large-pore mesoporous silica nanoparticles (LPMSNs) functionalized with amine groups (LPMSN-amine) for the delivery of yaluronici acid (HA)	- Increased HA loading and release capacity compared to standard contact lenses	- Likely only applicable for the delivery of HA, more testing on the delivery of other types of drugs is needed	Not currently in clinical use, research use only	[36]

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
