# Peer review of "Exploring the Potential of Nanoporous Materials for Advancing Ophthalmic Treatments"

_ijms, 2023, doi:10.3390/ijms242115599_

Round 1

Reviewer 1 Report

The review article was well-designed and well-organized; however, minor changes are required before publication.

1. The title of the review suggests Ophthalmic Treatments but why the in the review the main focus on contact lenses and intraocular lenses?

2.    In Table 1, references to a few studies are missing.

3. The graphical abstract needs to be revised, showing more of the sequence of review, not the proper outcome.

4.    Only one figure is presented in the review article.  More well-presented figures should be added for better understanding and attraction of the reader.

5.    In line 583, its mentioned that cyclodextrins have limits on total drug capacity, what about thiolated CDs, also being used in ocular treatment due to better mucoadhesive properties?

Minor changes are required before acceptance.

Author Response

Dear Reviewer,

We are exceedingly grateful for your insightful comments and the time you have invested in reviewing our manuscript. We are pleased that you found the review article to be well-designed and well-organized. Your constructive feedback has significantly contributed to improving the quality of our work. Below, we provide a detailed response to each of your comments:

  1. Title Clarification: Thank you for pointing out the perceived focus on contact lenses and intraocular lenses. Our review consists of three major parts: drug delivery systems (eye drops/injections), contact lenses, and intraocular lenses. Your suggestion to make this clearer is valuable. However, to maintain brevity in the title, we are considering keeping the same title.
  2. Missing References in Table 1: We apologize for this oversight. The missing references have now been included in Table 1 to provide a more comprehensive understanding of the topic at hand.
  3. Graphical Abstract Revision: Your input regarding the graphical abstract is well-taken. We have revised it to better represent the sequence and structure of the review.
  4. Inclusion of Additional Figures: As you rightly mentioned, the inclusion of additional figures would augment both the understanding and the engagement of the reader. In response, we have incorporated several new figures to provide a more thorough and visually appealing discussion of the subject matter.
  5. Thiolated Cyclodextrins in Ocular Treatment: We appreciate your keen observation about the capabilities of thiolated cyclodextrins (t-CDs) in drug delivery systems. We have amended the section to include the following: “Thiolated cyclodextrins (t-CDs) developed within a nanoparticulate intraocular lens (NP IOL) could also serve as an effective drug delivery system. While no IOLs have been developed with t-CDs to date, Kesavan et al. have amalgamated dexamethasone-embedded cyclodextrins within a Carbopol 980 NF and sodium carboxymethylcellulose hydrogel, demonstrating promising therapeutic results in a rabbit model [93]. Given that cyclodextrins improve the solubility of hydrophobic drugs [94] and t-CDs possess mucoadhesive properties that extend drug residence time on the ocular surface [95], this technology could potentially enhance ocular drug delivery from an IOL or contact lens drug depot.”

We trust that these revisions adequately address your concerns. Once again, we sincerely thank you for your constructive feedback, and we look forward to your further thoughts on the amended manuscript.

Reviewer 2 Report

The review article by Wu and collaborators deals with a very novel topic of nanomaterials used for advanced ophthalmic treatments. However, it seems that the ambitious goal was not fully realized by the authors. The article organization makes it hard to follow, and English language requires a thorough revision. In this form the article does not deserve publication. Below are my specific comments.

Major remarks:

1/ factual errors - lines 83-84 - the given sizes of mega- and microporous materials are wrong and have to be corrected

lines 87-90 - what are "organic elements"? Hydrogen, carbon, oxygen and nitrogen are the main elements of organic compounds but they cannot be termed as "organic" themselves. On the other hand, inorganic materials cannot be made from carbon compounds.

lines 233-234 - the pore sizes are wrong

2/ figure 1 - if it is a graphical abstract, it should not be embeded in the manuscript. Moreover, the numbers of Sections are wrong and do not overlap with the manuscript sections numbering.

3/ chapter 2.2 - it is very general, it would be better to elaborate on some physical, chemical and biological properties that make these materials useful for ophthalmic applications.

4/ chapter 3.1.1 and further - it makes no sense to create such short subchapters consisting of 2-3 sentences

5/ distribution of content between chapters - in my opinion it is sometimes confusing for the readers, e.g. lines 294-295 and 366-367 - the Authors write about using contact lenses for drug delivery and in chapter 5 again about CL for drug delivery. The same with intraocular lenses - lines 391 and further, and again in chapter 4.

Minor remarks:

1/ lines 44-55 - there are wrong numbers of sections in the text

2/ lines 596, 611, 618 - it is recommended to write the first author name and "et al." to cite multi-authored publications

3/ repetitions of information, e.g. in lines 574-575

4/ line 646 - wrong table number

English language requires a thorough revision due to grammar and wording errors

Author Response

Dear Reviewer,

Thank you for your comprehensive review and thoughtful remarks regarding our manuscript. We value your expertise and the effort you have put into identifying areas for improvement. Your critique is instrumental in elevating the quality of our work. Here, we present our point-by-point responses to your comments:

Major Remarks

  1. Factual Errors in Lines 83-84, 87-90, 233-234:
    • We appreciate your attention to factual inaccuracies, particularly those related to the pore sizes of materials. We have corrected this section to align with the International Union of Pure and Applied Chemistry (IUPAC) classification. It now reads: "According to the International Union of Pure and Applied Chemistry (IUPAC) classification, porous materials are separated into three main categories based on pore size: megaporous materials (>50 nm), mesoporous materials (2-50 nm), and microporous materials (<2 nm)."
    • Additionally, your observation about the mischaracterization of "organic elements" is noted and highly appreciated. Thank you for the valuable feedback – We’Ve taken your suggestions into consideration to clarify the usage of the term "organic elements" in the manuscript and made the appropriate changes to enhance the clarity. I understand that the main elements of organic compounds cannot be terms as organic themselves; therefore I have made the following revisions to address this:
      1. Original text: "Organic elements, such as hydrogen, carbon, oxygen, and nitrogen, play a crucial role in the formation of complex organic compounds"
      2. Revised text: "Elements, such as hydrogen, carbon, oxygen, and nitrogen, play a crucial role in the formation of complex organic compounds"
    • I've removed the word "organic" before "elements" to clarify that these specific elements themselves are not organic, but rather make up compounds that are terms as "organic". This modification ensures a more accurate representation of the terminology. Furthermore to maintain consistency, I have also made similar modifications to the sentence following it, which discusses inorganic compounds. I believe the revised text now accurately reflects the classification of organic and inorganic compounds. Moreover, the text in lines 87-90 has been revised to read: "Inorganic nanoporous materials are made from non-organic and pure metal-type materials, such as zeolites, silicates, ceramics, aluminum, and titanium[17]." This clarification should resolve the confusion.
    • Lastly, to address the concern about the clarity and structure of the language used, we have conducted three rounds of proofreading with different proofreaders to enhance coherence and eliminate any linguistic errors.
  2. Figure 1 and Section Numbering:
    • We appreciate your observation regarding Figure 1. It has now been provided as a separate figure and is not embedded within the manuscript. Additionally, we have rectified the section numbering to ensure consistency throughout the paper.
  3. Chapter 2.2 Elaboration:
    • Following your recommendation, we have enriched this section with insights on the physical, chemical, and biological properties of nanoporous materials that make them apt for ophthalmic applications. A summarized excerpt is: "Nanoporous materials have unique physiochemical and biological characteristics ideal for ocular drug delivery. They are non-toxic, biodegradable, customizable in terms of pore structure, and exhibit high surface area-to-volume ratios, facilitating optimized drug absorption and bioavailability. Thus, nanoporous materials promise to improve therapeutic outcomes while minimizing side effects."
  4. Short Subchapters in 3.1.1 and Further:
    • We acknowledge the inadequacy of short subchapters and have consolidated information where applicable, eliminating the fragmented presentation.
  5. Content Distribution:
    • We concur that the distribution of content between chapters, particularly regarding contact lenses and intraocular lenses for drug delivery, was confusing. This has been rectified, and the content has been realigned for better comprehension and flow.

Minor Remarks

  1. Section Numbers in Lines 44-55:
    • The section numbers have been corrected to match the updated content structure.
  2. Citation Style in Lines 596, 611, 618:
    • Thank you for this stylistic note. Citations have been updated to include the first author's name followed by "et al."
  3. Repetitions in Lines 574-575:
    • We apologize for the redundancy and have removed the repeated information for clarity.
  4. Table Numbering in Line 646:
    • The table number mentioned has been corrected.

We trust that these amendments sufficiently address your queries and concerns. We sincerely hope the revisions meet your approval and look forward to your continued guidance.

Best regards,

Round 2

Reviewer 2 Report

The Authors have responded to all my remarks and corrected their review article. These corrections improved the quality of manuscript considerably, however, I would still recommend some improvements before publications.

Specific remarks:

1- lines 44-55: numbering of sections should be unified throughout the manuscript, please, correct it to arabic numbers

2- lines 149 and 165: correct Figure X to consequent number

3- lines 431 and 438: I would suggest to remove the subchapters 5.1.1. and 5.1.2, and leave the text as a 5.1. chapter

4- line 639: conclusions should be numbered as chapter 7

5- what is the source of figures? if they are prepared based on some other publications, they must be cited in the figure caption

Minor editing of English is required

Author Response

Dear Reviewer,

We are most grateful for your continued engagement with our manuscript and for the additional comments aimed at enhancing its quality. We find your insights extremely helpful and are pleased to report that we have made further revisions in accordance with your latest suggestions:

  1. Lines 44-55: Numbering of sections
    We appreciate your suggestion to unify the numbering of sections throughout the manuscript. We have revised the numbering scheme to Arabic numbers as recommended, ensuring consistency.
  2. Lines 149 and 165: Correct Figure Numbers
    Thank you for pointing out the inconsistency in the figure numbering. We have carefully reviewed these lines and have corrected the figure numbers to maintain a consecutive order.
  3. Lines 431 and 438: Removal of Subchapters 5.1.1. and 5.1.2
    We concur with your suggestion that the subchapters 5.1.1. and 5.1.2 may not be necessary and could potentially disrupt the flow of the manuscript. We have therefore merged the content into a single 5.1 chapter to enhance readability and cohesiveness.
  1. Line 639: Numbering of Conclusions as Chapter 7
    Thank you for pointing out the oversight in the numbering. We sincerely apologize for our mistake in referencing the "Conclusions" section as Chapter 4 instead of 7. This error has been rectified in the revised manuscript.
  2. Source of Figures: We appreciate your concern regarding the source of the figures in the manuscript. Each figure is original work created for this article. To clarify this, we have now included a statement in the figure captions clarifying this.

We sincerely appreciate your time and expertise in reviewing our manuscript for a second time. We hope that our revisions have adequately addressed your concerns, and we look forward to any further feedback you may have.

Thank you once again for your valuable input.

Sincerely